computational chemistry/medicinal chemistry

mexidol, emoxypine, density functional theory study, antiradical activity, antioxidants

**Author for correspondence:**
Quan V. Vo
e-mail: vvquan@ute.udn.vn

# Theoretical insights into the antiradical activity and copper-catalysed oxidative damage of mexidol in the physiological environment

Nguyen Thi Hoa[1], Mai Van Bay[2], Adam Mechler[3] and Quan V. Vo[1]

[1]The University of Danang, University of Technology and Education, Danang 550000, Vietnam
[2]Department of Chemistry, The University of Danang, University of Science and Education, Danang 550000, Vietnam
[3]Department of Chemistry and Physics, La Trobe University, Victoria 3086, Australia

(iD) QVV, 0000-0001-7189-9584

Mexidol (**MD**, 2-ethyl-6-methyl-3-hydroxypyridine) is a registered therapeutic agent for the treatment of anxiety disorders. The chemical structure suggests that **MD** may also act as an antioxidant. In this study, the hydroperoxyl radical scavenging activity of **MD** was studied to establish baseline antioxidant activity, followed by an investigation of the effect of **MD** on the copper-catalysed oxidative damage in biological systems, using computational methods. It was found that **MD** exhibits moderate radical scavenging activity against HOO$^{\bullet}$ in water and pentyl ethanoate solvents following the single electron transfer and formal hydrogen transfer mechanisms, respectively. **MD** can chelate Cu(II), forming complexes that are much harder to reduce than free Cu(II): **MD** chelation completely quenches the Cu(II) reduction by ascorbic acid and suppresses the rate of reduction reaction by O$_2^{-}$ that are the main reductants of Cu(II) in biological environments. Therefore, **MD** exerts its anti-HO$^{\bullet}$ activity primarily as an OIL-1 inhibitor.

## 1. Introduction

Mexidol (**MD**, emoxypine, 2-ethyl-6-methyl-3-hydroxypyridine) is a drug used primarily for the treatment of anxiety. It is known to have antiischemic, antihypoxic, neuroprotective, antistress, nootropic and geroprotective properties [1,2]. **MD** is also used as an antioxidant for reducing tissue damage by reactive oxygen species [1,3]. For this reason, the radical

This article has been edited by the Royal Society of Chemistry, including the commissioning, peer review process and editorial aspects up to the point of acceptance.

mexidol (**MD**)

**Figure 1.** Molecular structure and atomic numbering of mexidol (**MD**).

scavenging activity of **MD** was assessed in experimental studies [3,4]. The rate constant of **MD** reaction with peroxyl radical in methyl oleate was measured as $k = 2.8 \times 10^4 \, M^{-1} \, s^{-1}$ [4], while that for 1,4-dioxane solvent was $k = 2.1 \pm 0.3 \times 10^4 \, M^{-1} \, s^{-1}$ [3]. However, organic solutions such as methyl oleate and 1,4-dioxane are not suitable model environments to assess *in vivo* activity; in the physiologically relevant aqueous environment **MD** may exhibit very different behaviour due to dissociation of the OH moiety. Thus there is an impetus to study the antiradical activity of **MD** in physiological media.

In evaluating the antioxidant activity of **MD**, one has to also consider that pyridines are very good coordinating ligands and therefore it is highly likely that **MD** forms chelates with trace metals [5,6]. Previous studies showed that the presence of transition metals such as Cu(I) can produce hydroxyl radicals via the Fenton-like reaction (1.1) [7–11]:

$$Cu^+ + H_2O_2 \rightarrow Cu^{2+} + HO^\bullet. \tag{1.1}$$

However, Cu(I) is not the most stable or abundant form of copper in physiological environments. The concentration of Cu(I) is defined by the amount of Cu(II) ions undergoing reduction by hyperoxide radical ($O_2^{\bullet-}$) or deprotonated ascobic acid ($AA^-$) with the rate constants $k = 4.46 \times 10^9 \, M^{-1} \, s^{-1}$ and $1.33 \times 10^8 \, M^{-1} \, s^{-1}$, respectively [10]:

$$Cu^{2+} + O_2^{\bullet-} \rightarrow Cu^+ + {}^3O_2 \tag{1.2}$$

and

$$Cu^{2+} + AA^- \rightarrow Cu^+ + AA^\bullet. \tag{1.3}$$

Chelation of Cu(II) has the potential to inhibit or suppress reactions (1.2) and (1.3) and therefore indirectly inhibit reaction (1.1). Thus, the capacity of **MD** to inhibit Cu(II) reduction by chelation is also of interest (figure 1).

Previous studies showed that computational method offers the most convenient path for studying structure–activity relationships in radical reactions to guide the design of novel antioxidants with enhanced activity [12–19]. In several prior studies, the radical scavenging activity of organic compounds in physiological environments was successfully evaluated by quantum chemistry calculations [20–23]. Thus, in this study, the HOO$^\bullet$ radical scavenging activity of **MD** was assessed in lipid and polar media using the quantum mechanics-based test for overall free radical scavenging activity (QM-ORSA) protocol [13,20]. Cu(II) chelation ability was also assessed and the ability of **MD** to act as an OIL-1 inhibitor of the copper-catalysed oxidative damage in biological systems was investigated.

# 2. Computational details

All density functional theory (DFT) calculations were carried out with Gaussian 09 suite of programs [24]. M06-2X functional [25] and 6-311++G(d,p) basis set were used for all calculations. The M06-2X functional is one of the most reliable methods to study thermodynamics and kinetics of radical reactions [20,25–30].

The stability of Cu(II) chelates was compared by calculating the Gibbs free energy of formation for all possible chelates that were first constructed with MD based on $[Cu(H_2O)_4]^{2+}$ geometry, optimized by molecular mechanics calculations using the Spartan software [31], then energy-minimized with DFT as per above.

The kinetic calculations were performed following the QM-ORSA protocol [13,20] and following the literature [29,30,32–37]. The details of the method are shown in electronic supplementary material, table S1. Atom-in-molecule (AIM) analysis [38] was performed by using the AIM2000 software [39].

# 3. Results and discussion

## 3.1. The HOO˙ radical scavenging activity of **MD**

### 3.1.1. Gas phase evaluation

To reduce computing time, the radical scavenging activity of **MD** was first evaluated in the gas phase following the liturature [16,40], according to the three main reaction pathways: formal hydrogen transfer (FHT), single electron transfer followed by proton transfer (SETPT) and sequential proton loss electron transfer (SPLET) [41,42]. Radical adduct formation (RAF) is another common pathway that however requires a localized C=C bond; the aromatic ring of **MD** cannot support this mechanism [21,43,44]. The probabilities of the three feasible antioxidant mechanisms (FHT, SETPT and SPLET) were first evaluated by computing the main thermodynamic parameters associated with these mechanisms: bond dissociation enthalpy (BDE), ionization energy (IE) and proton affinity (PA), respectively. The calculated BDE, IE and PA values of **MD** are shown in electronic supplementary material, table S3.

The lowest BDE value was calculated for O3−H at 84.4 kcal mol$^{-1}$. This value is higher than that of natural antioxidants such as resveratrol (83.9 kcal mol$^{-1}$) [45], piceatannol (73.1 [21] or 75.1 [46] kcal mol$^{-1}$), Trolox (73.0 kcal mol$^{-1}$) [47] and ascorbic acid (73.9 kcal mol$^{-1}$) [47]. The BDE values of the C−H bonds are even higher by about 2.0–16.0 kcal mol$^{-1}$. At the same time, the IE and PA values are about 2.3 and 4.1 times higher than the lowest BDE value. Thus, based on the computed data, the antioxidant activity of **MD** in apolar and low-dielectric environments is projected to proceed via the FHT pathway. Calculations of $\Delta G°$ (Gibbs free energy change) values of the defining step of the **MD** + HOO˙ reaction (electronic supplementary material, table S3) also confirmed that the FHT mechanism is the main path of the HOO˙ radical scavenging activity of **MD** in the gas phase as well as in non-polar media.

Based on the thermodynamic results, the kinetics of the antiradical activity of **MD** was evaluated for the thermodynamically favourable positions and mechanisms according to the QM-ORSA protocol [20], and the data are shown in electronic supplementary material, table S4, and figure 2. The results suggest that the O3−H bond ($\Delta G^{\neq} = 13.0$ kcal mol$^{-1}$; $k_{\mathrm{Eck}} = 2.95 \times 10^5$ M$^{-1}$ s$^{-1}$; $\Gamma = 100.0\%$) is the only feasible site for H-abstraction during the **MD** + HOO˙ reaction. The C−H bonds do not make any contribution ($\Gamma = 0\%$) in the overall rate constant of the antiradical activity of **MD**. That is consistent with the lowest BDE(O−H) values as calculated in the thermodynamic section. Thus the results suggest that the HOO˙ radical scavenging activity of **MD** is dominated by the FHT mechanism at the O3−H bond; therefore, this reaction is further analysed in lipid medium.

### 3.1.2. The HOO˙ radical scavenging activity of **MD** in physiological environments

In aqueous environment, the antiradical activity of acidic species is typically dominated by the ionic form [26,30,48]. Therefore, the protonation state of **MD** was first evaluated at physiological pH to find the most likely radical scavenging reactions. The **MD** structure allows protonation at the N1−H and O3−H bonds (figure 3); thus the p$K_a$ values of **MD** were calculated based on the literature [48,49] and are shown in figure 3.

The calculated p$K_a$ values for the amine were p$K_{a1} = 7.17$ (for N−H bond of cation form) and p$K_{a2} = 9.79$ (for O3−H bond). Therefore, in pH = 7.4 aqueous solution, **MD** exists in three states: the cation (H$_2$A$^+$, 37.0%), neutral (HA, 62.7%) and anion (A$^-$, 0.3%) states. Therefore, all three states were used in the kinetic evaluation of HOO˙ radical removal of **MD** in water at physiological pH = 7.4.

The preferred radical scavenging pathways of the neutral and anionic states have been established; the cationic state requires initial evaluation. The computing of $\Delta G°$ of the H$_2$A$^+$ + HOO˙ reaction for the possible pathways (electronic supplementary material, table S5) showed that the H$_2$A$^+$ + HOO˙ reaction is only clearly spontaneous for FHT at the C7−H bond ($\Delta G° = -4.4$ kcal mol$^{-1}$). Thus this reaction was used to calculate the rate constant of the HOO˙ radical scavenging of H$_2$A$^+$ in the aqueous solution. The overall rate constant of HOO˙ + **MD** reaction was computed following equations (3.1) and (3.2); the results are presented in table 1 and figure 2.

In the lipid medium

$$k_{\mathrm{overall}} = k_{\mathrm{app}}\,(\mathrm{FHT(O3-H)-neutral}). \tag{3.1}$$

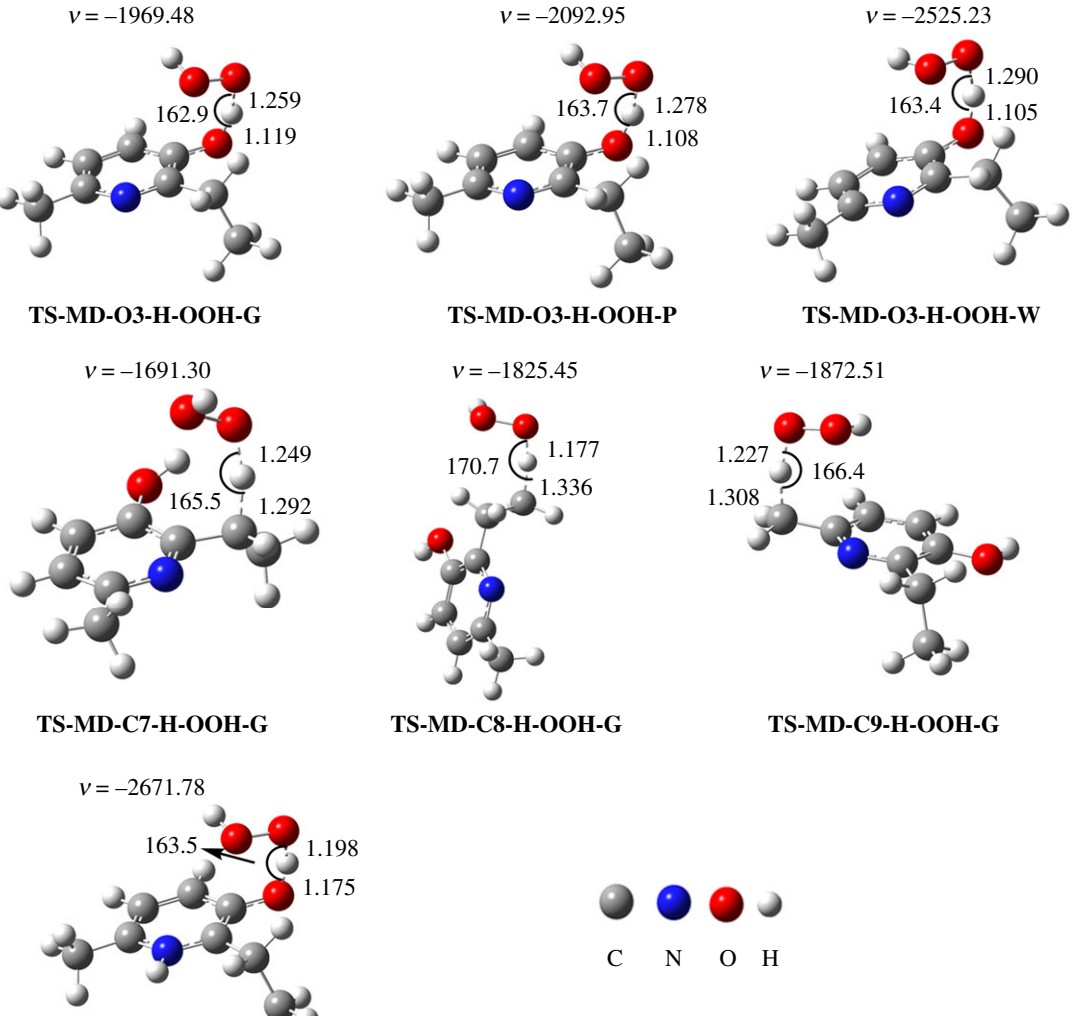

$v = -1969.48$  TS-MD-O3-H-OOH-G

$v = -2092.95$  TS-MD-O3-H-OOH-P

$v = -2525.23$  TS-MD-O3-H-OOH-W

$v = -1691.30$  TS-MD-C7-H-OOH-G

$v = -1825.45$  TS-MD-C8-H-OOH-G

$v = -1872.51$  TS-MD-C9-H-OOH-G

$v = -2671.78$  TS-MD-N1-H-O3-H-OOH-W

C    N    O    H

**Figure 2.** Optimized structures of the transition states (TS) along the FHT pathway of the HOO$^{•}$ radical scavenging activity of MD: (G) in the gas phase, (P) in pentyl ethanoate and (W) in water.

HO [⊕N–H] $pK_{a1} = 7.17$ / pH = 7.40 → HO [N] $pK_{a2} = 9.79$ / pH = 7.40 → [⊖O] [N]

37.0%        62.7%        0.3%

**Figure 3.** The deprotonation of **MD** at pH = 7.4.

In the aqueous solution

$$k_{\text{overall}} = k_{\text{f}}\,(\text{SET} - \text{HA}) + k_{\text{f}}\,(\text{SET} - \text{A}^{-}) + k_{\text{f}}\,(\text{FHT} - \text{H}_2\text{A}^{+}(\text{C7} - \text{H})) + k_{\text{f}}\,(\text{FHT} - \text{HA}(\text{O3} - \text{H})). \quad (3.2)$$

As per table 1, the HOO$^{•}$ + **MD** reaction in pentyl ethanoate is moderate with $k_{\text{overall}} = 4.40 \times 10^{3}\,\text{M}^{-1}\,\text{s}^{-1}$ by the FHT mechanism at the O3–H bond ($\Gamma = 100\%$). By contrast, **MD** exhibits good antiradical activity in water with $k_{\text{overall}} = 2.68 \times 10^{4}\,\text{M}^{-1}\,\text{s}^{-1}$. This reaction was defined ($\Gamma \sim 100\%$) by the SET pathway of the dianion A$^{-}$. The rate constant for the HA + HOO$^{•}$ reaction following the FHT mechanism at the O3–H bond is $k_{\text{f}} = 3.70 \times 10^{3}\,\text{M}^{-1}\,\text{s}^{-1}$ ($\Gamma = 13.8\%$), whereas that for the C7–H bond (H$_2$A$^{+}$) is only $k_{\text{f}} = 1.44 \times 10^{-3}$ M$^{-1}$ s$^{-1}$ and this reaction makes a negligible contribution (approx. 0%) to the total antiradical activity of **MD**. Compared with the reference antioxidant Trolox ($k = 1.30 \times 10^{5}$ and $1.00 \times 10^{5}\,\text{M}^{-1}\,\text{s}^{-1}$ in polar and

**Table 1.** Calculated $\Delta G^{\neq}$ (kcal mol$^{-1}$), $\kappa$, rate constants ($k_{app}$, $k_f$ and $k_{overall}$ (M$^{-1}$ s$^{-1}$)), and $\Gamma$ (%) in the **MD** + HOO˙ reaction in the studied media.

| mechanisms | | pentyl ethanoate | | | | water | | | | | |
|---|---|---|---|---|---|---|---|---|---|---|---|
| | | $\Delta G^{\neq}$ | $\kappa$ | $k_{app}$ | $\Gamma$ | $\Delta G^{\neq}$ | $\kappa$ | $k_{app}$ | $f$ | $k_f$ | $\Gamma$ |
| SET | HA | | | | | 36.8 | 18.1[a] | $4.80 \times 10^{-17}$ | 0.627 | $3.01 \times 10^{-17}$ | 0.0 |
| | A$^{-}$ | | | | | 5.1 | 16.0[a] | $7.70 \times 10^{6}$ | 0.003 | $2.31 \times 10^{4}$ | 86.2 |
| FHT | H$_2$A$^{+}$ | | | | | 24.4 | 483.1 | $3.90 \times 10^{-3}$ | 0.37 | $1.44 \times 10^{-3}$ | 0.0 |
| | HA | 15.9 | 328.1 | $4.80 \times 10^{3}$ | 100.0 | 16.6 | 1466.0 | $5.90 \times 10^{3}$ | 0.627 | $3.70 \times 10^{3}$ | 13.8 |
| $k_{overall}$ | | | | $4.80 \times 10^{3}$ | | | | | | $2.68 \times 10^{4}$ | |

[a]The nuclear reorganization energy ($\lambda$, kcal mol$^{-1}$); $f$ = %A$^{-}$/100; $k_f = f \cdot k_{app}$; $\Gamma = k_f \cdot 100/k_{overall}$.

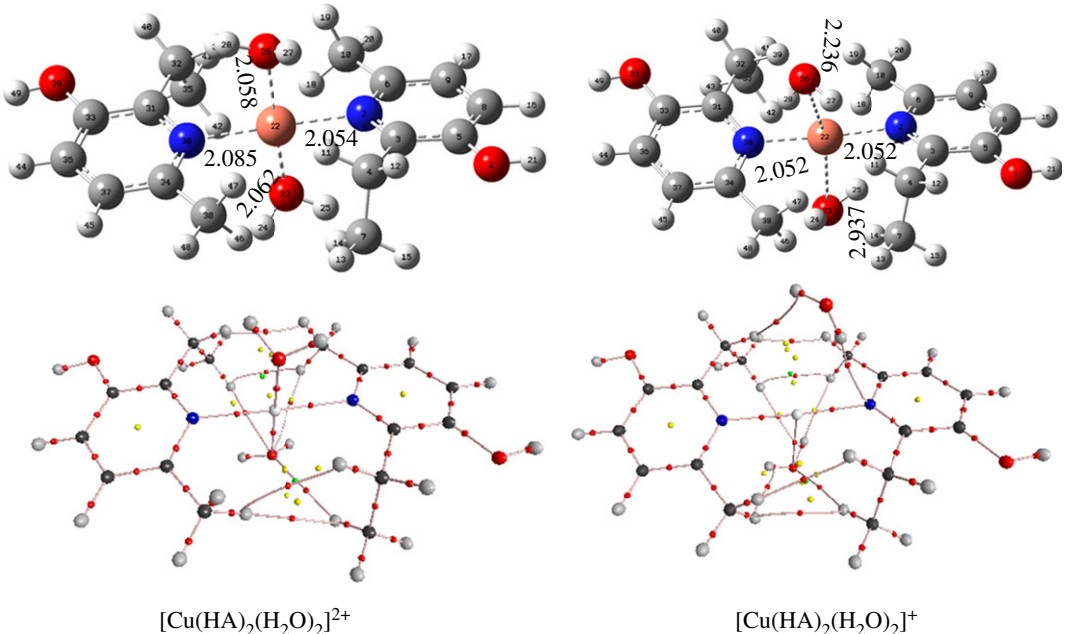

$[Cu(HA)_2(H_2O)_2]^{2+}$    $[Cu(HA)_2(H_2O)_2]^+$

**Figure 4.** Optimized structures and AIM topological shapes of $[Cu(HA)_2(H_2O)_2]^{2+}$ and $[Cu(HA)_2(H_2O)_2]^+$ in aqueous solution, at 298.15 K. Small red spheres indicate bond critical points.

**Table 2.** The calculated $\Delta G^\circ$, $\Delta G^{\neq}$, $\lambda$ (kcal mol$^{-1}$), and rate constants ($k_{app}$, $k_f$ and $k_{overall}$ (M$^{-1}$ s$^{-1}$)) at 298.15 K in the **MD** + Cu(II) reactions in water at pH = 7.40.

| reactions | $\Delta G^\circ$ | $\Delta G^{\#}$ | $\lambda$ | $k_{app}$ | $f$ | $k_f$ | $k_{overall}$ |
|---|---|---|---|---|---|---|---|
| $H_2A^+ + Cu^{2+}$ | 39.6 | 94.7 | 5.3 | $2.20 \times 10^{-57}$ | 0.37 | $8.14 \times 10^{-58}$ | $2.19 \times 10^7$ |
| $HA + Cu^{2+}$ | 20.5 | 28.7 | 6.2 | $6.00 \times 10^{-9}$ | 0.627 | $3.76 \times 10^{-9}$ | |
| $A^- + Cu^{2+}$ | −10.9 | 2.8 | 4.1 | $7.30 \times 10^9$ | 0.003 | $2.19 \times 10^7$ | |

nonpolar media, respectively) [40], the HOO$^\bullet$ radical scavenging activity of **MD** is lower in all of the studied environments. However, according to the empirical test for radical scavenging (activity should exceed $10^3$) [20,50] **MD** can be considered an antioxidant, albeit a weak one.

## 3.2. OIL-1 inhibition of copper-catalysed oxidative damage in biological systems

It was first calculated whether **MD** in any form can reduce free Cu(II), i.e. whether the chelation itself is reductive or not. Results are shown in table 2. Only the anionic form yields a negative Gibbs free energy change and thus it may reduce Cu(II). However, in the catalytic cycle, it competes with $O_2^{\bullet-}$ and $AA^-$ that have an order of magnitude higher rate constants ($4.46 \times 10^9$ M$^{-1}$ s$^{-1}$ and $1.33 \times 10^8$ M$^{-1}$ s$^{-1}$, respectively) [10]; hence **MD** does not contribute substantially to the reduction of free Cu(II).

To find the most stable complexes of the Cu(II) + **MD** reaction in water at pH = 7.4, the Cu(II) chelating ability of **MD** was evaluated considering the possible chelation sites of **MD** involving N and O atoms (electronic supplementary material, table S2). It was found that all of **MD** states ($H_2A^+$, HA and $A^-$) present in water could react with Cu(II) following 23 reactions and forming a range of complexes (electronic supplementary material, figure S1). The lowest Gibbs free energy of formation ($\Delta G^\circ$, kcal mol$^{-1}$) was predicted for the complex $[Cu(HA)_2(H_2O)_2]^{2+}$ (reaction 14, $N$ site, electronic supplementary material, table S2; figure 4) at −82.5 kcal mol$^{-1}$ that is more than two times lower than the second lowest value (reaction 8, $N$ site, electronic supplementary material, table S2). The equilibrium constant for reaction 14 is $K = 3.01 \times 10^{60}$ M$^{-1}$ and this chelate will make up 100% of the possible complexes, based on Maxwell–Boltzmann calculations. The mono-, tri- and tetra-coordinate Cu–**MD** complexes were also investigated; however, these reactions were not favoured for the **MD** +

**Table 3.** Selected parameters at the BCPs at intermolecular contacts for $[Cu(HA)_2(H_2O)_2]^{2+}$ and $[Cu(HA)_2(H_2O)_2]^{+}$ in aqueous solution, at 298.15 K.

| contacts | $\rho(r)$ (au) | $\nabla^2\rho(r)$ (au) | $G(r)^{a)}$ (au) | $V(r)^{b)}$ (au) | $G(r)/|V(r)|$ | $H(r)^{c)}$ (au) | $E_{HB}^{d)}$ (kcal mol$^{-1}$) |
|---|---|---|---|---|---|---|---|
| $[Cu(HA)_2(H_2O)_2]^{2+}$ | | | | | | | |
| N2 ⋯ Cu | 0.0695 | 0.3124 | 0.0872 | −0.0962 | 0.9057 | −0.0091 | −30.2 |
| N30 ⋯ Cu | 0.0745 | 0.3426 | 0.0959 | −0.1061 | 0.9035 | −0.0102 | −33.3 |
| O23 ⋯ Cu | 0.0597 | 0.3479 | 0.0878 | −0.0886 | 0.9908 | −0.0008 | −27.8 |
| O25 ⋯ Cu | 0.0603 | 0.3514 | 0.0887 | −0.0895 | 0.9907 | −0.0008 | −28.1 |
| $[Cu(HA)_2(H_2O)_2]^{+}$ | | | | | | | |
| N2 ⋯ Cu | 0.0753 | 0.3492 | 0.0999 | −0.1125 | 0.8881 | −0.0126 | −35.3 |
| N30 ⋯ Cu | 0.0753 | 0.3494 | 0.0999 | −0.1124 | 0.8886 | −0.0125 | −35.3 |
| O23 ⋯ Cu | 0.0112 | 0.0441 | 0.0101 | −0.0092 | 1.0991 | 0.0009 | −2.9 |

**Table 4.** Calculated $\Delta G°$, $\lambda$, $\Delta G^{\neq}$ (kcal mol$^{-1}$), and rate constants ($k_{app}$, $k_f$ and $k_{overall}$ (M$^{-1}$ s$^{-1}$)) for the reduction of Cu(II) and the most likely Cu(II) chelates by $O_2^{\bullet-}$ and AA$^{-}$ in aqueous solution, at 298.15 K. $k(Cu^{2+} + O_2^{\bullet-}) = 4.46 \times 10^9$ M$^{-1}$ s$^{-1}$; $k(Cu^{2+} + AA^{-}) = 1.33 \times 10^8$ M$^{-1}$ s$^{-1}$ [9,10].

| reactions | $\Delta G°$ | $\Delta G^{\neq}$ | $\lambda$ | $k_{app}$ | $f$ | $k_f$ | $k_{overall}$ |
|---|---|---|---|---|---|---|---|
| $[Cu(HA)_2(H_2O)_2]^{2+} + O_2^{\bullet-}$ | −10.5 | 4.6 | 36.2 | $2.10 \times 10^9$ | 0.9975[a] | $2.10 \times 10^9$ | $2.10 \times 10^9$ |
| $[Cu(HA)_2(H_2O)_2]^{2+} + AA^{-}$ | 14.6 | 16.9 | 31.5 | 2.80 | 0.9993[b] | 2.80 | 2.80 |

[a]$f(O_2^{\bullet-})$.
[b]$f(AA^{-})$.

Cu(II) in water as indicated by the higher values of $\Delta G°$ of formation, compared with the $[Cu(HA)_2(H_2O)_2]^{2+}$. Thus, the antioxidant activity calculations were performed for this chelate.

For a deeper understanding of the structure and stability of the complexes, AIM analysis was applied to $[Cu(HA)_2(H_2O)_2]^{2+}$ and $[Cu(HA)_2(H_2O)_2]^{+}$ in water. The topological shape and selected parameters at the bond critical points (BCPs) at metal–molecule contacts of the complexes are presented in figure 4 and table 3.

It can be observed that $[Cu(HA)_2(H_2O)_2]^{2+}$ is mainly stabilized by Cu(II) ⋯ O and Cu(II) ⋯ N contacts. That was affirmed by the existence of BCPs (red spheres) in the Cu(II)–O23, Cu(II)–O25, Cu(II)–N2 and Cu(II)–N30 electron densities. It can be inferred from table 3 that all of the Cu(II) ⋯ X (X = O, N) electron densities are partly covalent in nature as indicated by $\nabla^2\rho(r) > 0$, $G(r)/|V(r)| \leq 1$ and $H(r) \leq 0$ [51,52]. These contacts play a decisive role in the stability of the complex $[Cu(HA)_2(H_2O)_2]^{2+}$ with the significant negative values of $E_{HD}$ (−27.8 to −33.3 kcal mol$^{-1}$).

On the other hand, the stability of Cu(I) complex ($[Cu(HA)_2(H_2O)_2]^{+}$) is defined by the Cu(I) ⋯ N2(30) electron fluxes ($\nabla^2\rho(r) = 0.3492(0.3494)$, $G(r)/|V(r)| = 0.8881(0.8886)$, $H(r) = −0.0126(−0.0125)$ and $E_{HD} = −35.3$ kcal mol$^{-1}$; table 3). The contact of Cu(I) ⋯ O23 (H$_2$O) is a weak interaction with $\nabla^2\rho(r) = 0.0441$, $G(r)/|V(r)| = 1.0991$, $H(r) = 0.0009$ and $E_{HD} = −2.9$ kcal mol$^{-1}$. Thus two H$_2$O molecules in the $[Cu(HA)_2(H_2O)_2]^{+}$ are only solvating the system. That is consistent with previous studies of the coordination numbers and geometries of Cu(II) and Cu(I) complexes [9,53].

To evaluate the capacity of **MD** to reduce the copper-induced oxidative stress following the OIL-1 process in water at pH = 7.4, reduction reactions of the most stable complex ($[Cu(HA)_2(H_2O)_2]^{2+}$) and free Cu(II) were evaluated against typical copper-reducing species ($O_2^{\bullet-}$ and ascorbic acid anion (AA$^{-}$)) and the results are shown in table 4. It was found that the rate constant of the reaction of $O_2^{-}$ with $[Cu(HA)_2(H_2O)_2]^{2+}$ is $2.10 \times 10^9$ M$^{-1}$ s$^{-1}$ that is about 2.1 times lower than that of the reduction of free Cu(II) in water ($k(Cu^{2+} + O_2^{-}) = 4.46 \times 10^9$ M$^{-1}$ s$^{-1}$) [10]. However, complexation suppressed the rate constant of AA$^{-}$-driven reduction of Cu(II) by about $10^8$ times compared to free Cu(II). Based on the calculated data, **MD** is a good OIL-1 inhibitor of Cu(II)-catalysed oxidative stress.

# 4. Conclusion

The antioxidant activity of **MD** was investigated by evaluating the radical scavenging activity and the OIL-1 suppression of copper-catalysed oxidative damage in biological systems using computer calculations. It was found that **MD** exhibits moderate hydroperoxyl radical scavenging activity in both lipid and polar media. The antiradical activity in non-polar environments follows the FHT mechanism at the O3−H bond, whereas in aqueous solution, it follows the SET pathway of the anion state. Chelation with **MD** could suppress the Cu(II) reduction by $O_2^{\cdot-}$ and $AA^-$ in aqueous environment. Thus **MD** is predominantly an OIL-1 antioxidant.

Data accessibility. All relevant necessary data to reproduce all results in the paper are within the main text, electronic supplementary material and the Dryad Digital Repository: https://doi.org/10.5061/dryad.bzkh18997.

Authors' contributions. N.T.H.: data curation, formal analysis, investigation, validation, visualization, writing—original draft; M.V.B.: data curation, formal analysis, investigation, validation, writing—original draft; A.M.: conceptualization, project administration, software, supervision, writing—review & editing; Q.V.V.: conceptualization, formal analysis, methodology, project administration, resources, software, supervision, validation, writing—original draft, writing—review & editing. All authors gave final approval for publication and agreed to be held accountable for the work performed therein.

Competing interests. We declare we have no competing interests.

Funding. This research is funded by the Vietnamese Ministry of Education and Training under project no. B2021-DNA-16.

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
