## [Peer Review File · Royal Society Open Science]

Review History

RSOS-211239.R0 (Original submission)

Review form: Reviewer 1

Is the manuscript scientifically sound in its present form?

Yes

Are the interpretations and conclusions justified by the results?

Yes

Is the language acceptable?

Yes

Do you have any ethical concerns with this paper?

No

Have you any concerns about statistical analyses in this paper?

No

Recommendation?

Accept with minor revision (please list in comments)

Comments to the Author(s)

In this manuscript, the authors report a theoretical investigation on the primary and secondary antioxidant properties of Mexidol in gas-phase, water and lipid-like environments. The Cu chelating properties have been tested by using a 1:2 metal ligand ratio. The followed computational protocol is quite standard. The manuscript is suitable for the publication but the following points should be taken into account:

- the abstract is too long and must be reduced evidencing the obtained results;
- Figure caption are not clear;
- Gas phase results should be useful only for some comparison but are not necessary in the main text. They must be shifted in a SI section;
- The bibliography is incomplete and the following work must be quoted: Annu. Rev. Food Sci. Technol. 2016, 7:335-52; Food Chem. 2011, 125:288-306; J. Agric. Food Chem. 2006, 54, 6343-6351;
- some typos must be removed.

Review form: Reviewer 2**Is the manuscript scientifically sound in its present form?**

Yes

Are the interpretations and conclusions justified by the results?

No

Is the language acceptable?

Yes

Do you have any ethical concerns with this paper?

No

Have you any concerns about statistical analyses in this paper?

No

Recommendation?

Major revision is needed (please make suggestions in comments)

Comments to the Author(s)

Provide the data in Table S1 and Table S2 as mentioned in the text.

Decision letter (RSOS-211239.R0)

Dear Dr Vo:

Title: Theoretical insights into the antiradical activity and copper-catalyzed oxidative damage of mexidol in the physiological environment
Manuscript ID: RSOS-211239

Thank you for submitting the above manuscript to Royal Society Open Science. On behalf of the Editors and the Royal Society of Chemistry, I am pleased to inform you that your manuscript will be accepted for publication in Royal Society Open Science subject to minor revision in accordance with the referee suggestions. Please find the reviewers' comments at the end of this email.

The reviewers and handling editors have recommended publication, but also suggest some minor revisions to your manuscript. Therefore, I invite you to respond to the comments and revise your manuscript.

Please also include the following statements alongside the other end statements. As we cannot publish your manuscript without these end statements included, if you feel that a given heading is not relevant to your paper, please nevertheless include the heading and explicitly state that it is not relevant to your work. We have included a screenshot example of the end statements for reference.

- Ethics statement

Please clarify whether you received ethical approval from a local ethics committee to carry out your study. If so please include details of this, including the name of the committee that gave consent in a Research Ethics section after your main text. Please also clarify whether you received informed consent for the participants to participate in the study and state this in your Research Ethics section.

OR

Please clarify whether you obtained the necessary licences and approvals from your institutional animal ethics committee before conducting your research. Please provide details of these licences and approvals in an Animal Ethics section after your main text.

OR

Please clarify whether you obtained the appropriate permissions and licences to conduct the fieldwork detailed in your study. Please provide details of these in your methods section.

- Data accessibility

It is a condition of publication that you make available the data and research materials supporting the results in the article. Datasets should be deposited in an appropriate publicly available repository and details of the associated accession number, link or DOI to the datasets must be included in the Data Accessibility section of the article (<https://royalsocietypublishing.org/rsos/for-authors#question17>). Reference(s) to datasets should also be included in the reference list of the article with DOIs (where available).

Please include a Data Availability section after your main text stating where supporting data are available from, or where they will be made available should your article be accepted for publication.

<http://datadryad.org/submit?journalID=RSOS&manu=RSOS-211239>

- Competing interests

Please include a Competing Interests section after your main text declaring any financial or non-financial competing interests. If you have no competing interests please state 'I/we have no competing interests.'

- Authors' contributions

Please include an Authors' Contributions section at the end of your main text detailing the contribution of each author. All authors should have read and approved the manuscript before submission and this should be stated in the Authors' Contributions section.

The list of Authors should meet all of the following criteria; 1) substantial contributions to conception and design, or acquisition of data, or analysis and interpretation of data; 2) drafting the article or revising it critically for important intellectual content; and 3) final approval of the version to be published.

- Acknowledgements

- Funding statement

Please include a funding section after your main text which lists the source of funding for each author.

Because the schedule for publication is very tight, it is a condition of publication that you submit the revised version of your manuscript before 02-Dec-2021. Please note that the revision deadline will expire at 00.00am on this date. If you do not think you will be able to meet this date please let me know immediately.

- 1) A text file of the manuscript (tex, txt, rtf, docx or doc), references, tables (including captions) and figure captions. Do not upload a PDF as your "Main Document".
- 2) A separate electronic file of each figure (EPS or print-quality PDF preferred (either format should be produced directly from original creation package), or original software format)

- 3) Included a 100 word media summary of your paper when requested at submission. Please ensure you have entered correct contact details (email, institution and telephone) in your user account
- 4) Included the raw data to support the claims made in your paper. You can either include your data as electronic supplementary material or upload to a repository and include the relevant doi within your manuscript
- 5) All supplementary materials accompanying an accepted article will be treated as in their final form. Note that the Royal Society will neither edit nor typeset supplementary material and it will be hosted as provided. Please ensure that the supplementary material includes the paper details where possible (authors, article title, journal name).

Kind regards,
Dr Ellis Wilde
Publishing Editor, Journals

On behalf of the Subject Editor Professor Anthony Stace and the Associate Editor Dr Debashree Ghosh.

RSC Associate Editor

Comments to the Author:

The manuscript may be accepted after the authors revise the manuscript according to the referee comments and provide a point-wise reply.

RSC Subject Editor

Comments to the Author:

(There are no comments.)

Reviewer comments to Author:

Reviewer: 1

Comments to the Author(s)

In this manuscript, the authors report a theoretical investigation on the primary and secondary antioxidant properties of Mexidol in gas-phase, water and lipid-like environments. The Cu chelating properties have been tested by using a 1:2 metal ligand ratio. The followed

computational protocol is quite standard. The manuscript is suitable for the publication but the following points should be taken into account:

- the abstract is too long and must be reduced evidencing the obtained results;
- Figure caption are not clear;
- Gas phase results should be useful only for some comparison but are not necessary in the main text. They must be shifted in a SI section;
- The bibliography is incomplete and the following work must be quoted: Annu. Rev. Food Sci. Technol. 2016, 7:335–52; Food Chem. 2011, 125:288–306; J. Agric. Food Chem. 2006, 54, 6343-6351;
- some typos must be removed.

Reviewer: 2

Comments to the Author(s)

Provide the data in Table S1 and Table S2 as mentioned in the text.

Author's Response to Decision Letter for (RSOS-211239.R0)

See Appendix A.

Decision letter (RSOS-211239.R1)

Dear Dr Vo:

Title: Theoretical insights into the antiradical activity and copper-catalyzed oxidative damage of mexidol in the physiological environment
Manuscript ID: RSOS-211239.R1

It is a pleasure to accept your manuscript in its current form for publication in Royal Society Open Science. The chemistry content of Royal Society Open Science is published in collaboration with the Royal Society of Chemistry.

Yours sincerely,
Dr Ellis Wilde
Publishing Editor, Journals

On behalf of the Subject Editor Professor Anthony Stace and the Associate Editor Dr Debashree Ghosh.

RSC Associate Editor

Comments to the Author:

The authors have revised their manuscript as per the referee comments and have addressed all the issues satisfactorily. Therefore, the paper may be accepted.

Reviewer(s)' Comments to Author:

Appendix A

Dr Ellis Wilde

Publishing Editor, Journals

Royal Society of Chemistry

Thomas Graham House

Science Park, Milton Road

Cambridge, CB4 0WF

Dear Dr Ellis Wilde,

We have revised our manuscript meticulously following the reviewers' recommendations. Their comments allowed us to make several improvements to the manuscript. Please see the details in the response to reviewers file. Our responses are in blue and the changes are highlighted in red in the manuscript.

We believe that the manuscript is now ready for publishing.

Sincerely yours,

Quan Van Vo

The University of Danang - University of Technology and Education,

Danang 550000, Vietnam

Email: vvquan@ute.udn.vn

Danang, November 24, 2021

Reviewers' Comments to Author:

Reviewer comments to Author:

Reviewer: 1

Comments to the Author(s)

In this manuscript, the authors report a theoretical investigation on the primary and secondary antioxidant properties of Mexidol in gas-phase, water and lipid-like environments. The Cu chelating properties have been tested by using a 1:2 metal ligand ratio. The followed computational protocol is quite standard. The manuscript is suitable for the publication but the following points should be taken into account:

Author reply: We thank the reviewer for the recommendation.

-the abstract is too long and must be reduced evidencing the obtained results;

Author reply: The abstract has been rewritten.

-Figure caption are not clear;

Author reply: The figure captions have been revised.

-Gas phase results should be useful only for some comparison but are not necessary in the main text. They must be shifted in a SI section;

Author reply: As requested, most of gas phase data (Table 1, Table 2) have been moved to the SI file. Since the gas phase evaluation is an important part of the initial antioxidant screen in this study, the discussion was kept in the main text of manuscript. The thermodynamic data of the anion state (Table 3) has now been also moved to SI file (as Table S5).

-The bibliography is incomplete and the following work must be quoted: Annu. Rev.

Food Sci. Technol. 2016. 7:335–52; Food Chem. 2011, 125:288–306; J. Agric. Food Chem. 2006, 54, 6343-6351;

Author reply: The references have been added.

-some typos must be removed.

Author reply: The manuscript has been proofread and revised carefully.

Reviewer: 2

Comments to the Author(s)

Provide the data in Table S1 and Table S2 as mentioned in the text.

Author reply: The Table S1 is the methodology section thus all of the data have already been included. As for Table S2 (The Cartesian coordinates and energies of all of the complexes) data have now been moved and updated in Table S6.